# Multi-Task Deep Learning: Simultaneous Segmentation and Survival Analysis via Cox Proportional Hazards Regression

## Abstract

Multi-task learning has taken an important place as a tool for medical image analysis, namely for the development of predictive models of disease. This study aims at developing a new *deep learning* model for simultaneous segmentation and survival regression, using a version of the Cox model to support the learning process. We use a combination of a 2D U-net and a residual network to minimize a combined loss function for segmentation and survival regression. To validate our method, we created a simple synthetic data set - the model segments circles of different sizes and regresses the area of circles. The main motivation of this work is to create a workflow for segmentation and regression for medical images application: in specific, we use this model to segment lesions or organs and regress clinical outcomes as overall or disease-free survival.

**Keywords:** Multi-task learning; Survival analysis; Cox regression; Segmentation

## 1. Introduction

"Multi-task learning (MTL) is an approach to inductive transfer that improves learning for one task by using the information contained in the training signals of other related tasks" (Caruana, 1997). MTL has been shown to be helpful when access to data is limited (Zhang and Yang, 2018). Predicting clinical outcomes from medical images has been one of the main focus of research, as the power of prognosis is of great importance. Classically, Cox proportional hazard (CPH) models for survival prediction have been used to describe the effects of observed covariates on the risk of an event occurring (e.g. death) (Cox, 1972). Deep convolutional neural networks (CNNs) (LeCun et al., 1998; Krizhevsky et al., 2012) have achieved remarkable success across several imaging analysis tasks. Specifically, deep learning (LeCun et al., 2015) has achieved state-of-the-art results in object recognition and image segmentation. The adaptation of the CPH models to CNNs for survival prediction have been recently proposed, including the Deepsurv and Cox-nnet methods (Ching et al., 2018; Katzman et al., 2018).

This work proposes a new method that simultaneously segments an object and regresses its area, using a deep-learning CPH-inspired model, directing the application of this method to survival prediction.

## 2. Architecture

We use three different network architectures to identify the role of MTL for both circle segmentation and area regression, and assess which would have the best performance. Figure 1 A, B, and C depict the three different architectures that, for the sake of simplicity, will be termed R-net, SR-net, and SthenR-net, respectively.

The R-net consists of a series of convolution blocks with residual connections (He et al., 2015), max pooling layers, and terminating with two fully connected layers. Both SR-net and SthenR-net have in its core the same structure as the original U-net (Ronneberger et al., 2015), with minor modifications. In the SR-net, to allow for regression, a max pooling function was added, followed by two fully connected layers at the end of the contracting path of the U-net. The SthenR-net terminates with the same structure as R-net, but the input to this part of the network is the feature map produced just before the last convolution layer of the U-net.

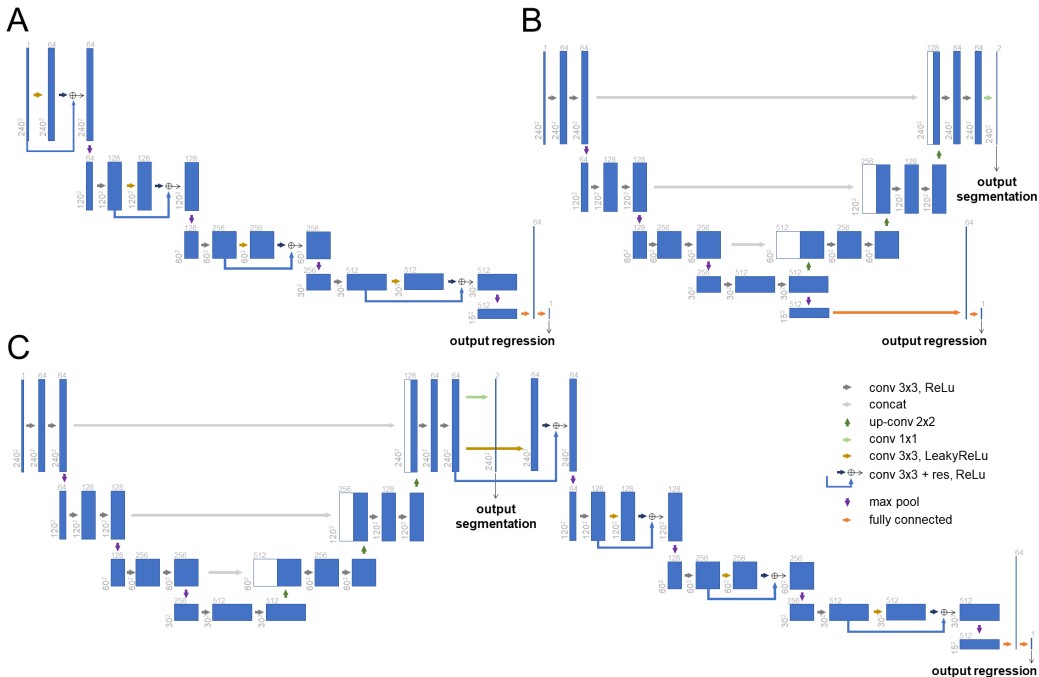

Figure 1: Networks architectures. A: R-net; B: SR-net; C: SthenR-net

## 3. Training

The synthetic data used was comprised of 500 images of size 240x240, each with a randomly sized and randomly located disk, contaminated with additive white Gaussian noise. For the regression task, we saved the area of each circle (min=4,max=11) and added a random amount (normal distribution mean=0 std=0.1). For our experiments, the dataset was divided into 80% training set (400 cases) and 20% test set (100 cases). To train the networks to perform segmentation and regression, the loss function used was a weighted sum of two different losses. For the segmentation, we used the loss function proposed by Isensee et al. (2018): a combination of dice and cross entropy losses. The regression loss was the partial log-likelihood, as used by Ching et al. (2018) and Katzman et al. (2018). These two losses were simply added with equal weights according to $\mathcal{L} = 0.5 * \mathcal{L}_{seg} + 0.5 * \mathcal{L}_{regress}$.

We used the Adam optimizer with learning rate $10^{-5}$. Additionally, to prevent overfitting, L2 regularization was used with decay $10^{-6}$ on both losses, and drop-out before the last fully connected layer was used keeping 50% of the neurons.

## 4. Experiments

Performance was measured with dice score for the segmentation task and with the concordance index between the ground truth and the predicted risk for the regression.

## 5. Results

At the end of training, both losses converged and did not present overfiting. Inference was calculated on both datasets (training and test) and the results are presented on Table 1. The Kaplan-Meier curves for each predictive model are presented on Figure 2.

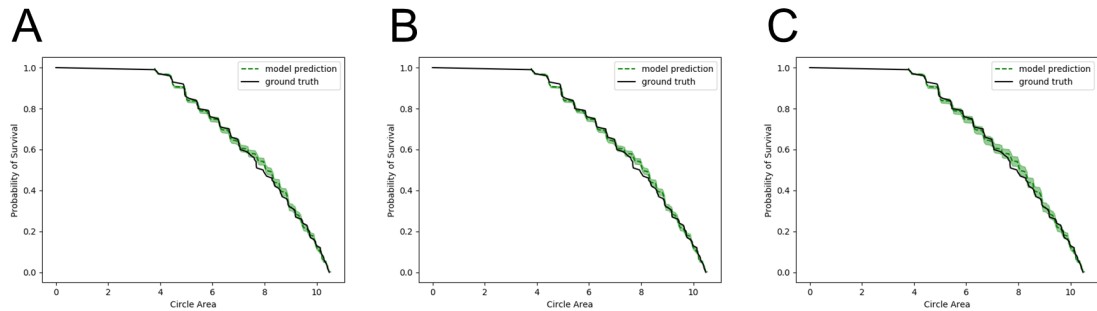

Figure 2: Kaplan-Meier curves for the test set. In black is the ground-truth and in green is the mean of the prediction (shade indicates the standard deviation). A: R-net; B: SR-net; C: SthenR-net.

## 6. Discussion

This work presented a new method for multi-task learning that aims to increase the performance of the regression task, with the help of the features extracted for the segmentation task. It serves as proof of concept for further work using medical imaging data and clinical outcomes from patients. Although these results require further validation (specifically cross-validation), they indicate that MTL can increase performance of the regression task. The combination of the two losses made the model learn both tasks. Furthermore, these results also suggest that to use the feature maps from the U-net bottleneck can help the performance of both tasks (better perfomance by SR-net against SthenR-net). The last layer just before the segmentation output seems not so informative as the feature map produced by the repeated convolutions and poolings of the original image. Further work includes the application of this method to real patient data.

Table 1: Performance of each network on the training (in gray) and test (in black) sets

| Method | Dice | | C-index | |
|---|---|---|---|---|
| | Training | Testing | Training | Testing |
| R-net | - | - | 0.904 | 0.908 |
| SR-net | 0.985+-0.021 | **0.980+-0.026** | 0.922 | **0.919** |
| SthenR-net | 0.965+-0.035 | 0.966+-0.0306 | 0.900 | 0.896 |

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
