# OpenReview forum: "Multi-Task Deep Learning: Simultaneous Segmentation and Survival Analysis via Cox Proportional Hazards Regression"
_MIDL.io/2020/Conference — Submitted to MIDL 2020_

### Official Review · AnonReviewer2 · 2020-03-09
**well-structured paper but missing important details**

**Rating:** 2
**Confidence:** 5

**Review:**

the paper is well-structured ane easy-to-follow, still I think important details are missing:
- the motivation of using a CPH model instead of other modelling of non-linear variable interactions,
- evaluation of more realistic cases, as the synthetic task is perhaps too simple
- statistical significance tests via cross validation
- some attention-based module to show that lesion localisation really helps the regression

---

### Official Review · AnonReviewer3 · 2020-03-11
**Interesting work but need to work on real data**

**Rating:** 2
**Confidence:** 5

**Review:**

The authors proposed a multi-task learning for simultaneous segmentation and survival analysis. The authors compared three architectures and validated that SR-Net can have best performance. Overall, it is a very interesting work as so far not much work investigate how segmentation and survival analysis can be trained together.

1. It is a pity that the authors only conducted experiments on the synthetic data. The proposed model has a very good potential use on real CT data for both segmentation and survival prediction. I am very curious about how the model work in practice.

2. It is not clear about how to use partial log-likelihood regression loss in your study. As we know, Cox Proportional Hazards Regression is designed to handle time-to-event predictions with censoring. However, in your study, the loss is used to regress the area of each circle. Why not use other regression loss like rmse ? How did you assign censoring label to your sample ?

---

### Official Review · AnonReviewer1 · 2020-03-11
**An interesting project but a limited abstract that requires a better pilot experiment**

**Rating:** 1
**Confidence:** 5

**Review:**

The authors propose a multi-task architecture to simultaneously perform image segmentation and survival analysis. The authors investigated three different architectures to analyse how the learned representations affect the performance across tasks. An experiment was performed to a synthetic dataset consisting of a randomly located disc in an image corrupted by Gaussian noise. The segmentation task was segmentation of the disc and the survival analysis/regression task was the disc area. The authors illustrate on this pilot experiment that the SR net architecture performed best, demonstrating that intermediate representations used regression achieved best performance.

[Strengths]
* This is an interesting problem, which is worthy of study. Multi-task learning in itself can be thought of as a representation learning problem and studying how to learn representations from medical images that enable accurate survival analysis is important.
* The idea to investigate which representations improve the regression task is interesting

[Weaknesses]
* For a synthetic dataset, I think the tasks are too simple and easy and it is difficult to draw any conclusions from this. The regression task is directly correlated to the segmentation task and could just be evaluated directly from the segmentation.
* Generally in multi-task problems, the performance on single-task networks is also included to showcase the benefit of jointly learning tasks. How good was the regression on its own?
* Why is the Cox-nnet loss function even needed in this scenario? Standard regression loss functions should have been compared to the partial log-likelihood of Ching et al.
* It is disappointing that there were not any experiments on medical data despite being a pilot study. Furthermore, the methodology is not novel enough to warrant such a simplistic synthetic experiment. As far as I can see, the novelty lies only in using a segmentation task in an multi-task setting to help learn representations for survival analysis.

[Suggestions/Further questions]
* Performance metrics on the training set (Table 1) are not needed
* Being a synthetic experiment with infinite data, was the training set divided into train/test?
* What does probability of survival have in relation to circle area?
* The regression task was circle area. How can this be used a substitution for risk prediction?

---

### Official Review · AnonReviewer4 · 2020-03-14
**Multi-Task Deep Learning: Simultaneous Segmentation and Survival Analysis via Cox Proportional Hazards Regression**

**Rating:** 1
**Confidence:** 3

**Review:**

Summary:- This paper develops new deep learning model for simultaneous segmentation and survival regression using Cox method which is a combination of 2D U-net and a residual network. Results are evaluated on synthetic data which model segments circles of varied sizes. This paper opens up about combining workflow for segmentation and regression  in medical imaging.

Strengths:- This paper proposed a way to simultaneously segments an object and regress it using CPH-inspired deep learning model. Paper has potential to survival prediction application.

Weaknesses:- Paper validation seems to be week. Synthetic data should be as close as possible to the real data. In general, gaussian noise is a too simplistic assumption for any realistic data. Introduction is quite week with no citations of recent work in survival prediction applications.

Major comments:-
Paper is very weekly written with the consideration of very simplistic synthetic dataset.
Figures quality needs to be improved.
More details about CPH model is required.
What is the batch size being considered during training? Also, please mention about inference timing.
Results and discussion sections need to be elaborated further.

---

### Meta-Review · Area_Chair1 · 2020-03-28
**MetaReview of Paper86 by AreaChair1**

**Rating:** 1

**Metareview:**

The authors propose a multi-task architecture to simultaneously perform image segmentation and survival analysis. The interest of the problem has been acknowledged by the reviewers.

The major flaw of the paper, as noted by all reviewers, is the experimental part: results are obtained on a synthetic dataset only (circle randomly located in the image), which appears to be too simple w.r.t. the real problem at hand. Also it is not clear how the synthetic regression problem, ie predicting the circle area, is related to the probability of survival.

The other point concerns the regression loss: it is unclear why a partial log-likelihood loss function is used, whereas standard regression losses (mse, mae…) could be assessed in the first place.

In addition to reviewing these two points,  I suggest the authors to enhance the paper by (i) highlighting clearly the performance of the single task network, (ii) performing statistical analysis to show the superiority of the proposed architecture, (iii) improving the quality of Kaplan-Meir curves in Figure 2.

**Paper Type:**

methodological development

---

### Decision · Program_Chairs · 2020-04-11

Reject